# Applying the Socioecological Model to Map Factors Associated with Military Physical Activity Adherence

**DOI:** 10.3390/ijerph20116047

**Published:** 2023-06-04

**Authors:** Golan Benisti, Orna Baron-Epel

**Affiliations:** 1Peres Academic Center, Rehovot 7610202, Israel; 2School of Public Health, Faculty of Social Welfare and Health Sciences, University of Haifa, Haifa 3498838, Israel; ornaepel@research.haifa.ac.il

**Keywords:** physical activity, barriers, motives, military, intentions, knowledge, survey, IDF, social norms, physical environment

## Abstract

Physical activity (PA) within the military can have large effects on the soldier’s health, productivity, and ability to meet tasks. This study aims to identify the factors associated with PA adherence during military service, applying the socioecological model, which classifies the factors influencing health behaviors into individual, social, and environmental levels. This cross-sectional survey was carried out among 500 soldiers aged 18 to 49 years in the Israeli Defense Forces. Statistical analysis to assess associations between PA and individual, social, and environmental factors included correlations, variance analyses, and multivariable linear regression. PA rates were higher among men soldiers in combat positions. Individual level factors, such as intention to perform PA (β = 0.42, *p* < 0.001), and self-efficacy regarding PA (β = 0.20, *p* < 0.001) were associated with PA among men and women. However, social norms were associated with PA only among men (β = 0.24, *p* < 0.001). The physical environment was not associated with PA adherence (β = −0.04, *p* = 0.210). Conclusions: Developing interventions on the individual level for all military personnel and interventions on the social level, mainly for men, could help increase levels of PA in the military.

## 1. Introduction

Physical inactivity is a major risk factor for morbidity and related mortality worldwide. The rates of physical inactivity are rising in many countries every year [1]. According to the World Health Organization’s (WHO) reports, the population mortality rate attributed to Physical Activity (PA) is approximately 6%, after hypertension (13%), smoking (9%), and diabetes (6%) [2]. In addition to the health implications of PA, physical inactivity increases the economic burden on a state’s health systems [3]. Currently, the updated recommendations for adults aged 18–64 years is to perform PA at least three times a week and to accumulate at least 150 min per week of moderate-intensity, or at least 75 min of vigorous-intensity aerobic activity, or a combination of the above. In addition, the American College of Sports (ACSM) and WHO recommended performing at least two resistance training sessions per week, which include all the large muscle groups and prioritizing multi-joint exercises [1,4].

The ecological model characterizes the factors related to behavior in general and the adoption of particular health behaviors. Ecological models are used as a framework for understanding the interactions between the individual and the environment, including the physical, social, and organizational environments. In other words, ecological models explicitly include psychological, environmental, social, and policy factors that are expected to influence behavior and incorporate a wide range of influences at multiple levels. In this study, we adopted the socioecological model as a theoretical basis for analyzing our hypothesized associations. Recently, there is a growing interest in ecological models due to their success in health promotion [5].

For example, socioecological models were found to be the most effective approach for tobacco control, and other health behaviors [6]. Other socioecological-based intervention programs included promoting healthy eating in school [7] and promoting PA habits during the COVID-19 pandemic [8].

Sallis et al. (1998) developed an ecological model for PA. In their model, the various factors are divided into influence levels, individual characteristics (biological and psychological), characteristics of the social and cultural environment, characteristics of the physical environment (natural and built), and policy (laws and regulations) [5]. Many studies tried to elucidate the motives for PA adherence and suggested that the main individual motives for PA are self-efficacy [9,10,11,12], enjoyment [13], the intention to lose weight, fitness, and body shape [14,15] where the main individual barriers were lack of enjoyment and lack of time [16,17,18], and laziness/lack of motivation [14,15].

Among the social and cultural motives was an encouraging social environment [5,18,19]. Some physical environmental factors included access to sports facilities, the existence of parks, and walking and biking trails [20,21].

According to an Organization for Economic Co-operation and Development (OECD) report, about 66% of adults across 23 OECD countries meet the recommended guidelines for moderate physical activity. Moreover, men are more likely to be physically active than women in all 23 OECD countries with comparable data [22]. 

In Israel, according to a national health survey published in 2019, about 35% of Israelis aged 21 years and older reported that they exercise at least three times a week for 30 min [15]. A higher percentage of men compared to women (41% vs. 29.6% respectively) met the recommendations, and this tends to be higher in older age groups [15]. 

Military service is a unique setting, which may affect and design the soldiers’ personality, attitudes, and behaviors, due to its special nature as a hierarchical organization. In particular, military service may influence health behaviors, including PA [23].

PA among soldiers serving in the military is a crucial element influencing many areas. Besides its influence on health and the economy, soldiers’ PA may affect organizational aspects of the military such as its ability to meet tasks and the soldiers’ productivity. Hence, PA in military organizations is performed in order to prepare the soldiers for their tasks during operational activity, combat, and routine [23,24]. According to the U.S. Army, the cost of a sedentary lifestyle and obesity to the US army is estimated to be 1.1 billion dollars per year [25]. However, despite the standards set by the U.S. Ministry of Defense, American soldiers are still exposed to the risk of a sedentary lifestyle and being overweight, since the military environment does not always support regular PA. In addition, soldiers are exposed to unhealthy food (such as energy-dense food), which is more accessible and convenient to consume, especially during operational activity [23]. 

In a study examining PA in the U.S. Army, it was found that the perceived advantages (e.g., weight management, reduced risk of chronic disease, increased lifespan, stronger physical abilities) and barriers (such as lack of time and lack of motivation) for PA, and self-efficacy were the main factors predicting participation of soldiers in PA. Social support was not found to be a predictive factor of PA [26].

Another study that examined health and PA habits in the U.S. Army found that special unit soldiers were more likely to be healthy and physically active compared to infantry unit soldiers. This difference may exist, due to more positive attitudes and higher social support regarding PA among special unit soldiers [27]. In this novel study, a meta-analysis was conducted, where the PA of different armies around the world was examined. Our study shows the various factors that influence the response to PA, and accordingly which interventions may be more effective in changing the health behavior of individuals. To the best of our knowledge, there has not yet been a study performed on any army that looks at the application of the ecological model of the military framework while also using a holistic view of the issue of PA. Our model tests this approach on the Israeli Defense Forces (IDF) army. 

A meta-analysis examining 136 studies regarding PA adherence in military organizations worldwide shows that PA interventions may be more effective when they are delivered by specialists such as fitness instructors and use theoretical basis or theory-based behavior change techniques (e.g., self-monitoring and goal-setting) [28].

The aim of this study is to identify motives and barriers for the adherence of PA at the individual, social, and environmental levels during military service in the IDF, by applying the socioecological model.

## 2. Materials and Methods

This study is a cross-sectional study stratified by military units of the IDF.

### 2.1. Participants

The survey was carried out among 500 soldiers and officers in mandatory as well as standing service (career soldiers) of the IDF. The study included 304 men and 196 women, the age range was between 18 and 49 years. The sampling framework was a stratified random sample based on a list of IDF units, from which 30 units were sampled randomly by type. 

Stratification was based on the various types of units—combat, mixed units (units that involve men and women soldiers), war support, and home front units, representing their relative size in the IDF. In each sampled unit, soldiers were randomly sampled. The interviewer visited the sampled units and randomly sampled soldiers (men and women) from each unit. The interviewer was present when the soldiers filled out a paper and pencil questionnaire in order to answer questions if they had difficulty answering the questionnaire. Soldiers were briefed beforehand about the study, including the details of the aim of the study, their right to refuse participation, privacy protection, and the average time it would take to fill out the questionnaire. 

This sample size was determined assuming a sampling error of 4.5% and a confidence interval of 95%. For a multiple regression with a low effect size of 0.05, with a power of 0.85, and α = 0.05, which includes up to 15 predictors, the desired sample size was calculated to be 450 participants [29,30].

### 2.2. Variables and Research Tools Questionnaire

The questionnaire was conducted by a researcher, a non-authority figure who was unknown to the soldiers (dressed in civilian clothes). The response rate was 96%, and the 4% of the soldiers who refused participation did so out of lack of time. The data collection in the study was anonymous and no personal details were collected from those who refused to participate. 

To reduce self-report bias in the measurements, the answers were validated. For instance, the weight of every 15th participant was measured and validated with the self-reported value.

### 2.3. Dependent Variable

PA was assessed using two questions from the WHO Global PA Questionnaire (GPAQ) [31]: (1) How many times do you perform PA in a manner that causes you to breathe heavily or to sweat? (1—never, 2—less than once a month, 3—once a month, 4—once a week, 5—2–3 times a week, 6—4–6 times a week, 7—every day); (2) During the last seven days, how many days did you perform PA, which lasted, in total, at least 30 min per day? (0—0 days, 1—one day, 2—two days, 3—three days, 4—four days, 5—five days, 6—six days, 7—seven days). Since these two items taken from a validated questionnaire represent similar content, and since Spearman’s correlation between them was high (r = 0.84, *p* < 0.001), they were unified into one dependent variable. Due to the different scales of both items, each item was standardized in the following manner: the score of each participant was calculated as the difference between the crude score and the sample’s average, divided by the sample’s standard deviation, such that the sample’s average became 0 and the standard deviation became 1. The dependent variable was defined as the average of the scores of both items, where a higher score indicates a higher frequency of PA.

### 2.4. Independent Variables Individual-Level Variables

Knowledge was defined as the soldier’s familiarity with the weekly recommended accumulated duration for PA. This was measured using a question from the MABAT survey of 2014–2016. MABAT is a series of national cross-sectional studies on the status of health and nutrition of the Israeli population, carried out on a representative sample of the population by age groups [22]. The participant was asked whether he/she knows what is the recommended weekly duration for PA, where the value 1 represents a correct answer (150 min), and the value 0 represents a wrong answer.

Intention to perform PA was assessed using a question taken from the WHO questionnaire on PA [31]. The participant was asked about his/her intentions to perform PA, using a scale of 1–4, where 1—inactive and not interested in PA, 2—inactive but interested in PA, 3—active in a satisfactory manner and does not wish to be more active, 4—active and wishes to be more active [31].

Self-efficacy was assessed using a questionnaire developed by Saunders [32]. The questionnaire contains five items, in which the participant is asked about his/her perception of his/her skills and ability to perform PA (“I am capable of performing PA even if I am busy most of the week”). The scale ranged between 1 to 5, where 1 indicated not capable at all, and 5 indicated highly capable. The Cronbach alpha was 0.84. The overall score was the average of all the items, where a higher score indicated higher self-efficacy. 

### 2.5. Social-Level Variables

Social norms were measured using 31 items taken from two questionnaires. The first one is Fishbein and Ajzen’s questionnaire (17 items) for norms and attitudes regarding PA [33] in which the participant is asked about perceptions and behaviors regarding PA acceptable in the military unit (“aware of the importance of PA”; “encouraging their friends to perform PA”). The second one is Zohar’s organizational safety climate questionnaire, which includes 14 items [34], in which the participant is asked about expectations, activities, and procedures related to PA in the unit (“in my unit the commanders allow free time for PA” etc.). 

### 2.6. Physical Environment-Level Variable

The physical environment was assessed using a list of six items, including various types of facilities for PA which may exist in the military unit (for example running trails, fields for ball games, a gym, etc.). The participants marked the facilities which were accessible to them in their unit. The overall score was built as the sum of the number of facilities reported. The score ranged between 0 and 6.

### 2.7. Background Variables

Data were collected for the following background variables: age, sex, Body Mass Index (BMI) [kg/m^2^] (calculated from the participant’s self-reported weight and height), self-reported number of sleeping hours per 24 h, position (combat/war support/home front), type of military service (standing/mandatory), and unit (special forces/infantry/field units/mixed war units/home front units). 

### 2.8. Data Analysis

Data processing was performed using SPSS version 27 software. Internal consistencies (Cronbach alpha) were calculated for the study’s variables, and compound variables were built. The soldiers’ PA levels were described, and the differences in PA by position, gender, and type of military service were evaluated using one-way ANOVA. Pearson’s correlation was used for evaluating the associations between PA level and the continuous background variables (such as age and BMI). The study associations were tested using a three-step stepwise multivariable linear regression. The first step included the variables gender, position, and the type of military service that were entered into the linear regression model as control variables. The second step included the variables of knowledge, intentions, self-efficacy, social norms, and physical environment. In the third step of the regression, we used the “simple slopes” method for identifying interactions between the independent study variables and gender, position, and type of service [35,36]. The participants’ age distribution was found to be positively skewed to the younger ages, and therefore a logarithmic transformation was applied to the age variable. All other continuous variables had normal distributions.

## 3. Results

Table 1 describes the participants’ background variables. About 60% of the 500 participants were men. The participants’ ages ranged from 18 to 49 years, and the average was 23.6 years. Only 13% had an academic degree. In addition, 61.4% of the participants were in standing service, and the rest were in mandatory service. The average BMI was about 24.5 kg/m^2^, and 75.6% of the participants reported good or very good health (self-reported health).

Table 2 describes the rate of PA adherence among the study participants.

PA at least once a week was reported by 42% of the participants. To validate the average for accuracy, the PA of individuals was reported for the previous week of study participation. During the last week, only 16.2% of the participants reported that they performed PA at least once a week, and only 4.2% of the participants reported adherence to PA at least four times during the last week.

The average PA adherence was found to be higher among combat soldiers compared to war support and home front units’ soldiers (*p* < 0.001). Moreover, PA averages were higher among men compared to women (*p* < 0.001). In addition, the PA level was higher among soldiers in mandatory service compared to soldiers in standing service. This could be due to the fact that soldiers in mandatory service are younger, as presented later. 

Negative correlations were found between age, BMI, and PA (r = −0.29, r = −0.16, *p* < 0.001, respectively), and a positive correlation was found between self-reported health and PA level (r = 0.39, *p* < 0.001), i.e., the higher the participants’ PA level the better the participants’ self-reported health. Moreover, the lower the participants’ age, BMI, and the better the participants’ self-reported health the higher their PA level. No association was found between the number of sleeping hours per 24 h and adherence to PA.

PA was positively associated with some individual variables, including an intention to perform PA and self-efficacy, i.e., higher intentions to perform PA and higher self-efficacy regarding PA are related to higher reported PA. Social norms, a social environment level of influence in the socioecological model, were also positively associated with PA, i.e., social norms encouraging PA were related to higher reported PA. No association was apparent between the physical environment and PA (Table 3).

We applied a multivariable linear regression, between the dependent variable, PA adherence, and the independent variables in order to assess their contribution to the variability in PA adherence (Table 4). 

Due to the correlations between BMI and age (r = 0.28, *p* < 0.001), and BMI and gender (*p* < 0.001), BMI was not included in the regression (Table 3 and Table 4). In addition, due to the correlation between position and military unit (*p* < 0.001), these variables were also not included in the regression (Table 3 and Table 4). Since most of the participants (76%) reported good or very good health, self-reported health was also not included in the regression (Table 3 and Table 4). Therefore, only gender, age, position, and type of military service were included in the regression.

Individual-level variables contributed 26% to the variance in PA adherence, which included intentions to perform PA (β = 0.42, *p* < 0.001) and self-efficacy (β = 0.20, *p* < 0.001) as predictors of PA according to our regression model.

In addition, the social-level variable, social norms (β = 0.24, *p* < 0.001), contributed 7% to the variance. 

Finally, the contribution of the background variable to the variance was 22%. In total, the study’s independent variables explained 55% of the variability in PA adherence. 

After adjustments, the physical environment variable was not found to be a statistically significant explanatory factor for PA. 

A single interaction between sex and social norms was found to be statistically significant (B = 0.275, SE = 0.07, β = 0.23, *p* < 0.001), and contributed 1.4% to the variability in PA level (*p* < 0.001). Interpreting the interaction using the “simple slopes” method (Aiken & West, 1991; Dawson, 2014) shows a positive association between social norms and PA level among men (coefficient = 0.32, t = 7.41′ *p* < 0.001), but not among women (coefficient = 0.05, t = 0.84, *p* = 0.402), i.e., the association between social norms and PA level exists solely among men.

## 4. Discussion

In this study, we divided the factors influencing PA in the military by the levels of the socioecological model and demonstrated that two of the levels are associated with PA in the military; individual and social levels (however, the physical environment level is not associated with PA). The military is characterized by the terms of its hierarchy of command, a rigidly stratified society, and a self-contained social world. As such, understanding the unique characteristics of the military may contribute to organizations with similar structures [37,38]. The reason we see a relationship between intentions/self-efficacy and PA came up clearly in our study. PA habits in the military come from the intention to exercise. This is consistent with what has been described in the literature, where a connection is reported between the individual’s positive mindset towards PA and performing it, according to Sallis and Owen. Another variable that was observed to be critical for PA, was the sense of self-efficacy. This means that believing one can reach a goal that was set for themselves is crucial in being able to meet the goal. This again is consistent with the literature which describes that those that are persistent in PA have a high perception of self-efficacy and vice versa (Peers, 2020). Other studies show that the perception of self-efficacy positively affects the frequency and participation in physical activity as well as the ability to deal with barriers related to physical activity [9,10].

Our study suggests that the main individual variables associated with PA adherence in the military are intention to perform PA and self-efficacy. Social environment, represented by the social norms variable, was also associated with PA only among men, but not among women. Therefore, PA adherence of women in the military is a function of individual-level factors, whereas PA adherence of men in the military is a function of individual-level factors as well as social level. A possible explanation for this difference between men and women may be that the military environment tends to be a “masculine environment” [39], and as such, the physical training sessions are aimed primarily at the desires and characteristics of men. Therefore, men may be more affected by the social norms related to PA in the military than women. 

As expected, the rate of soldiers meeting the recommendations for PA was higher among young men soldiers who serve in combat units. The gender differences reflected in our results comply with the existing literature. Lower levels of activity among women compared to men are frequent worldwide. This gender inequality is reduced when aspects of the built environment, such as walkability, are improved [40]. A possible explanation for the increased PA among combat unit soldiers is that combat soldiers may be more motivated compared to other soldiers since their acceptance to such units requires meeting strict physical demands. Physical training is essential for meeting these demands. In addition, combat unit soldiers may perceive their bodies as a tool for performing their tasks and meeting the physical demands of combat service. It may also be due to more positive attitudes and higher social support regarding PA among combat unit soldiers [27]. A selection bias may also explain this higher rate of PA among combat unit soldiers since combat units recruit soldiers with better health and better physical capabilities [41,42]. 

Our results demonstrate that the individual’s intention to perform PA is a significant factor in the adherence to PA of soldiers. It is known in the literature that psychological variables have a central role in a person’s intention to choose a specific behavior. Many studies suggest a strong association in the general population between an individual’s intention to perform PA and the PA performed de facto by the individual. A systematic review of 89 studies about intentions to perform PA and motivation regarding PA concluded that the association between intentions to perform PA and adherence to PA is statistically significant [43]. Our results strengthen the literature related to psychological variables and their role in an individual’s specific behavior. Therefore, increasing the intention to perform PA among soldiers may be a useful strategy for promoting PA adherence. Creating social norms which support and encourage PA in the military unit may create a positive change in the individual’s intention to perform PA. 

Another significant individual-level factor that we found to be associated with the adherence to PA by soldiers is self-efficacy. Self-efficacy is a central psychological variable that affects PA adherence. Previous studies have demonstrated the association between self-efficacy and PA adherence in the general population, as well as in specific populations such as school students and sports professionals [9,10,44]. Researchers reported that those who persist in PA have also a higher self-efficacy and vice versa [11,12]. Our results also strengthen these findings and widen the existing sparse knowledge about this association among soldiers. Performing PA continuously has a crucial impact on health [1]. Our results show that the self-efficacy of a soldier is directly associated with the soldier’s PA adherence during military service. Strengthening self-efficacy among soldiers may increase the rates of PA adherence during military service. Such strengthening may be done through enjoyable training sessions, which create a sense of achievement, success, and effectiveness.

On the other hand, some studies show that self-efficacy does not necessarily affect PA adherence in all cases. For example, a study shows that self-efficacy has a larger effect when dealing with PA which lasts for a long period of time, or with high-intensity PA [45]. Another study claims that self-efficacy can predict PA adherence, only if the PA is limited to a specific period of time. However, when the PA requires persistence for a longer period of time, it is doubtful whether it can be predicted by self-efficacy without using other variables. The authors suggest that self-efficacy is not able to solely explain PA adherence [46]. Our study deals with PA in accordance with the general recommendations, and not necessarily with intense PA or PA limited to a specific period of time, and therefore does not fall into the categories described in the above studies. In addition, in our study self-efficacy is not an independent potential predictor of PA but combined with other variables according to the social-ecological model.

It is evident in the literature that knowledge may have an indirect effect on the individual’s health behavior, by affecting the individual’s perceptions, and by that affecting the individual’s health behavior, as well as the individual’s willingness to adopt new behaviors [47]. At the same time, other studies claim that the effect of knowledge on the individual’s health behavior is small, and therefore the individual’s health behaviors cannot be predicted by knowledge without the support of other environmental variables [48]. Our results support these studies, as they show no significant association between the soldier’s knowledge regarding PA and self-reported PA.

Our results demonstrate that social norms of PA within the unit—a social-level factor—are directly associated with adherence to PA. The more the individual perceives PA as an important value of the individual’s social environment, the higher the probability that this individual will perform PA [5]. However, our results suggest that this association holds only for men soldiers and not for women soldiers. Hence, the PA of women in military service is a function of individual-level variables, and the PA of men in military service is a function of individual-level variables, as well as social-level variables. 

According to the literature, positive social norms regarding PA may lead to PA adherence, and negative social norms may reduce PA adherence. In other words, in places where the individual’s formal and informal surrounding social environment encourages and supports PA, the adherence rate of PA may be higher, and in places where the individual’s surrounding social environment does not encourage and support PA, the adherence rate of PA may be lower [16,19].

Commanders have a critical role in creating environments and opportunities for PA in the military unit. Commanders are able, using their personal example, to act as leaders, to enforce the military’s regulations regarding PA, to adapt them to the social norms, and to legitimate PA patterns in the military unit. Social norms, in every organization, begin with and are induced by the organization’s management [49]. Particularly, in the military, which is an organization whose management is the commanders, better social norms may be created by the military commanders. 

Gould, who is considered one of the leading figures in sport psychology research, claims that among the most important factors contributing to PA adherence is a supportive social environment [18]. This claim is confirmed by many other studies [50]. Our results are consistent with previous studies but also shed light on the interaction between gender and social norms with respect to PA adherence. 

Natural and built physical environments, which support and encourage PA, are considered a part of PA promotion strategy among health organizations worldwide and in Israel as well [16,51]. The literature emphasizes that physical environment characteristics are important for encouraging PA, but only if they are combined as a part of a comprehensive strategy, which involves the behavioral change of the individual [21]. Our results show no contribution of the physical environment to PA adherence in the military. These findings emphasize the conflict in the association between the physical environment and PA, which are independently associated with one another, but depend on the mediation of psychosocial factors. 

### Limitations

This study has some limitations. 

First, this is a cross-sectional study, and as such, it does not take into account the impact of processes occurring in the Israeli army during recent years on PA adherence among soldiers. In addition, causality cannot be concluded from a cross-sectional study. The study of health habits is known as an area in which inverse causality is frequent. Therefore, further longitudinal studies are required to establish a valid causality.

Second, our study has some potential reporting biases, since data collection in our study was based on self-reporting. We applied some procedures in order to reduce these potential biases. For example, we validated the soldiers’ reported body mass in the following manner: 30 participants out of the selected participants, 50% men and 50% women, were randomly sampled, and their body mass was measured by the interviewer to validate their reported body mass. 

In addition, a potential bias may exist in reporting PA adherence. However, according to scientific literature, self-reporting of PA is considered a valid tool for assessing PA adherence. For example, a study that examined 17 validated questionnaires for assessing PA adherence pointed out that self-reporting of PA had high validity, especially among young adults and middle-aged adults. In addition, it was mentioned that self-reporting was the most convenient tool for assessing PA adherence in national surveys [52].

## 5. Conclusions

The social-ecological model can help to understand the factors associated with PA. PA among women is a function only of individual-level variables, whereas PA among men is a function of individual-level variables, as well as the social environment level. The physical environment level seems not to have an impact on soldiers’ PA.

This study proposes that PA adherence during military service is better when the soldiers have a higher intention to perform PA, when their self-efficacy is high, and when the social norms of PA within the unit are high.

There is a need for further studies, such as longitudinal studies or trials, to establish the causality of the associations found in this study.

In addition, we recommend examining why women’s PA adherence is associated only with individual-level factors, whereas men’s PA adherence is associated with individual-level, as well as social-level factors.

Military service, which is mandatory in Israel, provides a unique opportunity for interventions among most of Israel’s population in order to promote healthy habits in general and PA adherence in particular.

Based on our study, it is recommended that various intervention strategies are examined, which may shape the desired health behavior in the military as well as in the general population by developing interventions on the individual level for all. Meanwhile, focusing on social factors (mainly for men) could help increase levels of PA in the military.

## Figures and Tables

**Table 1 ijerph-20-06047-t001:** Participants’ Background Variables.

Background Variable		*N*	%
Sex	Men	304	60.8
	Women	196	39.2
Age *M* (*SD*)	18–48	23.61	5.85
BMI, *M* (*SD*)	17–44	24.47	4.07
Country of origins	Israel	448	89.6
	Other	52	10.4
Marital status	Bachelor	416	83.2
	Married	80	16.0
	Divorced	4	0.80
Children	Yes	67	13.4
Religious level	Secular	281	56.2
	Conservative	146	29.2
	Religious	59	11.8
	Other	14	2.8
Educations	None-high school	47	9.4
	High school	373	74.6
	Academic	66	13.2
	Other	14	2.8
Type of service	Mandatory	193	38.6
	Standing	307	61.4
Position	Combat	141	28.2
	War support	179	35.8
	Home front	180	36.0
Military unit	Special unit	77	15.4
	Infantry	88	17.6
	Field unit	88	17.6
	War unit	65	13.0
	Home front unit	182	36.4
Self-reported health	Bad	24	4.8
	Medium	98	19.6
	Well/very well	378	75.6

**Table 2 ijerph-20-06047-t002:** Distribution of Physical Activity Adherence (N = 500).

Physical Activity General	Less Than Once a Month/Never N (%)	About Once a Month N (%)	1–3 Times a Week N (%)	At Least 4 Times a Week N (%)
M = 3.69 SD = 1.63 (Range 1–7)	130 (26.0%)	92 (18.4%)	211 (42.2%)	67 (13.4%)
Physical activity in the last week	Not at all	1–2 days	3–4 days	At least five days
M = 1.18 SD = 1.57 Range 0–7	257 (51.4%)	141 (28.2%)	81 (16.2%)	21 (4.2%)

**Table 3 ijerph-20-06047-t003:** Averages, Standard Deviations, and Pearson’s Coefficients between the Study Variables. (2: Age, 3: Knowledge regarding PA, 4: Intentions to exercise, 5: Sense of self-sufficiency, 6: Norms and social climate, 7: Physical environment).

Variable	M (SD)	2	3	4	5	6	7
1. Physical activity (standardized) (−1.20–+2.87)	0 (0.95)	−0.18 ***	0.13	0.64 ***	0.48 ***	0.50 ***	0.18 ***
2. Age	23.61 (5.85)		−0.01	−0.09	−0.06	−0.05	−0.11
3. Knowledge (0–1)	0.27 (0.45)			0.08	0.05	0.10	0.06
4. Intentions to perform PA (1–4)	2.89 (1.03)				0.42 ***	0.35 ***	0.16 ***
5. Self-efficacy (1–5)	3.56 (0.86)					0.30 ***	0.12
6. Social norms (1–5)	2.73 (0.94)						0.42 ***
7. Physical environment (0–6)	2.67 (1.55)						

Note. Bonferroni correction for multiple comparisons was applied (*p* = 0.002). *** *p* < 0.001.

**Table 4 ijerph-20-06047-t004:** Multiple Linear Regression of Physical Activity by Background Variables, Personal, Social, and Physical Variables.

Variable	Model 1		Model 2	
	β	*p*	β	*p*
Gender—Men	0.12	0.006	0.01	0.684
Age	−0.09	0.067	−0.08	0.048
Position—combat	0.39	<0.001	0.12	0.002
Position—home front	0.01	0.850	0.01	0.695
Type of service—standing	−0.02	0.672	−0.03	0.370
knowledge			0.06	0.068
Intentions to perform PA			0.42	<0.001
Self-efficacy			0.20	<0.001
Social norms			0.24	<0.001
Physical environment			−0.04	0.210
Adj. *R*^2^	0.22, *p* < 0.001		0.55, *p* < 0.001	

## Data Availability

Due to the nature of this research, participants of this study did not agree for their data to be shared publicly, so supporting data is not available.

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
