# Peer review of "Applying the Socioecological Model to Map Factors Associated with Military Physical Activity Adherence"

_ijerph, 2023, doi:10.3390/ijerph20116047_

Round 1

Reviewer 1 Report

Dear Authors,

Congratulations on this work. This study provides a good overview of the factors that influence soldiers’ physical activity behaviours; however, I do have some recommended changes needed to strengthen your manuscript. I have included my suggestions below. Wishing you all the best with the revisions!

Line 28: adding in “physical activity (PA)” so the acronym is clear to those who didn’t see the abstract.

Line 30: Whose recommendation? A lot of countries have adopted these suggestions, but always good to specify.

Line 60: are you talking about the Organisation for Economic Co-operation and Development? Best to add this before the acronym.

Line 81: what do you mean by perceived advantages? Expand.

Line 92: Add a justification for why this study matters. What are you adding that other studies missed?

Lines 94-95: switch so the full name is first and the acronym is in brackets.

Line 94: Is there a reason why you are excluding the policy level of the socio-ecological model from the analysis? As mentioned, the army has PA policies in place. I would recommend a justification for why you are excluding policy.

Lines 109: What was the consent process? What was the protocol if military personnel did not want to participate in the study?

Line 114: To clarify your methods, I would recommend adding a section at the beginning with a brief overview of the questionnaire (ex. how it was developed, how it was administered, and how long it took to complete) before getting into each of the variables.

Lines 170-172: This sentence seems more appropriate in the data analysis section.

Line 177: specify one-way or two-way ANOVA.

Line 187: Report how many participants you had in your study. Did you meet the sample size required? If you did not, justify why the findings are still valuable.

Lines 212-213: Where can we find this information on the relationship between sleep and PA? It doesn’t appear in any of the tables. It also looks like BMI is missing from the Pearson correlation. Make sure you have all of the variables you started with in the methods reported in your models.

Lines 263-264: Add a sentence saying why we see this relationship between intentions/self-efficacy and PA or how this fits in the literature (or combine it with the section you have added later in the discussion).

Line 290: Combine this paragraph with the next paragraph.

Line 296: Expand on what you mean by the literature. Add something like “strengthen the literature related to…”

Lines 333-339: I can see how you are trying to connect this paragraph with the next, but it is not clear how this fits in the manuscript. I would recommend moving this paragraph later in the manuscript (before the paragraph starting at line 354), so the reader knows what was found in your study and then gains the context as to why this happens.

Line 362: Combine with the previous paragraph.

Discussion section: when possible, try to use references specific to the military population. Findings from other populations (ex. older populations) might not apply to military personnel.

Lines 405-407: This is a general statement. I would suggest specifying your recommendations. You can combine it with the final sentence.

Reviewer 2 Report

Dear authors,

Firstly, I would like to congratulate you on the work done. This study aims to identify the factors associated with PA performance during military service using a socioecological model. Regarding the manuscript, I found it very interesting and is, in my humble opinion, worth publishing with a few corrections/ remarks.

 The introduction is well written in general, with only a few corrections to be made. In this regard, my main concern is with the phrase “PA performance”. You use this throughout the text and I think it is not exactly what you are studying. I believe that you mean “PA adherence” or “frequency of practice”, is it not? In my opinion, these phrases better reflect the aim of your work, as "performance" is more related to the level of proficiency that one person has while practising PA. If you agree on this, please change it in the whole manuscript.

 In line 45, you write: “… socioecological models were found to be a most effective approaches for tobacco control.” Please correct it to “… a most effective approach for…”

 Line 74-75: “According to the U.S. army, the 74 cost of sedentary lifestyle and obesity to the US army is estimated by 1.1 billion dollars 75 per year.” Please add reference

The methods are well written. I would only suggest that you write the procedures you took to reduce the bias in the self-reported measures.

Regarding the results, the way tables are presented is quite confusing, especially table 2. I think this one needs to be remade. For example, it seems that you place two headers in the table (first and third line), which is odd.

Table 3 has numbered headers but I cannot understand to what they refer. Can you clarify this?

Finally, can you better clarify what “other” education level is, in table 1?

The discussion seems appropriate and just needs minor tweaks. For example, Line 295:  - “Our results strengthen the knowledge exists in the literature.” Please correct English language

Once again, congratulations on the job well done, I hope my remarks help you improve it.

Round 2

Reviewer 1 Report

Dear authors,

Thank you for the opportunity to review your manuscript. It was great to read your manuscript again. You have done a good job integrating the recommendations from the reviewers. I only have two minor comments:

Line 97: Move up your description for the acronym “Israeli Defense Forces (IDF)” here.

Line 589: the reference style looks off. Update so it is the correct referencing style to the rest of the paper.

Besides those points, I have no additional changes to recommend. Congratulations on this work!
